# Molecular Evidence Reveals the Sympatric Distribution of *Cervus nippon yakushimae* and *Cervus nippon taiouanus* on Jeju Island, South Korea

**DOI:** 10.3390/ani12080998

**Published:** 2022-04-12

**Authors:** Maniram Banjade, Seon-Mi Park, Pradeep Adhikari, Sang-Hyun Han, Young-Hun Jeong, Jun-Won Lee, Sung-Hwan Choi, Hong An Nguyen, Hong-Shik Oh

**Affiliations:** 1Practical Translational Research Center, Jeju National University, Jeju-si 63243, Korea; mani88zoo@gmail.com (M.B.); psm0624@naver.com (S.-M.P.); 2Institute of Humanities and Ecological Consensus Laboratory, Hankyong National University, Anseong 17579, Korea; pdp2042@gmail.com; 3Species Restoration Technology Institute, Korea National Park Service, Yeongju 36015, Korea; hansh04@naver.com; 4Interdisciplinary Graduate Program in Advanced Convergence Technology and Science, Jeju National University, Jeju-si 63243, Korea; jyh6156@naver.com (Y.-H.J.); sung_1220@naver.com (S.-H.C.); nguyenhongan161@gmail.com (H.A.N.); 5Faculty of Science Education, Jeju National University, Jeju-si 63243, Korea; dlwnsdnjs70s@naver.com

**Keywords:** *CytB*, Jeju Island, phylogenetic study, sika deer, non-native species

## Abstract

**Simple Summary:**

The sika deer (*Cervus nippon*) was introduced in South Korea from Japan and Taiwan for commercial farming. They have become invasive to mainland South Korea and Jeju Island due to escape from confinement and illegal release. Native species and ecosystems may be threatened by the presence of non-native species. To deal with ecological risk and evolutionary processes, information on the phylogeny of these non-native sika deer is necessary. Genetic studies using mitochondrial DNA cytochrome B (*CytB*) gene sequences were conducted to determine the subspecies of Jeju sika deer and their phylogenetic relationship. On Jeju Island, we confirmed the presence of two distinct groups of *CytB* haplotypes: *Cervus nippon yakushimae*, native sika to Japan, and *Cervus nippon taiouanus*, native to Taiwan.

**Abstract:**

Non-native species threaten native ecosystems and species, particularly on islands where rates of endemism and vulnerability to threats are high. Understanding species invasion will aid in providing insights into ecological and evolutionary processes. To identify the non-native sika deer (*Cervus nippon*) population in Jeju, South Korea, and their phylogenetic affinities, we collected tissue samples from roadkill and the World Natural Heritage Headquarters in Jeju. Mitochondrial DNA cytochrome B (*CytB*) gene sequences were analyzed to determine two distinct *CytB* haplotypes. Phylogenetic analysis using maximum likelihood *tree* revealed two haplotypes of *CytB* clustered into two different groups representing two subspecies: *C. n. yakushimae*, native to Japan, and *C. n. taiouanus*, native to Taiwan. The tentative divergence time between the two subspecies was estimated at 1.81 million years. Our study confirmed that the two subspecies of sika deer are sympatric in the natural ecosystem of Jeju Island. This study provides valuable information to help government and conservation agencies understand alien species and determine control policies for conserving native biodiversity in South Korea.

## 1. Introduction

Native biodiversity, agriculture, and ecosystem services are all threatened by the introduction of non-native species [1,2,3]. Introduced alien plants and animals such as birds, reptiles, and mammals can alter the structure and function of the ecosystem, resulting in a decline in the population of native species [4,5]. The impacts of non-native species have typically been evaluated as more severe on islands than on continents [5]. Recently, 40% of the native species present on islands are at high extinction risk [6]. Therefore, the introduction of species into small islands and consequent problems caused by non-native species in natural ecosystems worldwide is concerning [7,8]. Herbivores such as deer, rabbits, and cattle can influence native island vegetation and cause various ecological impacts, including habitat loss and changing ecosystem dynamics [9].

Mammalian species are introduced accidentally [10,11] or intentionally [12] outside their native region. Following global practice, after the end of the Korean War, South Korea allowed the introduction of commercial farming of mammalian species such as sika deer, red deer, wild boar, and nutria [13,14]. The sika deer, native to East Asia, was introduced to South Korea from Taiwan and Japan [15]. Because of the escape from farmland and illegal release by farmers, the sika deer has been established as a wild population on Jeju Island, the largest southernmost island of the Korean Peninsula. Currently, the population of sika deer is rapidly growing and invading livestock farmlands and wild ecosystems. Therefore, the Ministry of Environment and the Jeju Especial Self-governing Province began controlling alien species population to limit their existence in the wild [16]. To control the population of non-native species efficiently, collecting basic biological information, including taxonomy and population distribution, is necessary. Therefore, the precise identification of non-native species and their origin is essential for determining control policies [17,18]. 

Mitochondrial DNA (mtDNA), because of its maternal inheritance, absence of recombinant events, and high mutation rate, has been used for studying genetic divergence among or within species [19,20]. In particular, the mtDNA of the cytochrome B gene (*CytB*), one of the most widely used molecular marker in animals [21], is used in resolving evolutionary relationships and taxonomic status of sympatric mammals globally [22,23,24]. In addition, the *CytB* gene sequences are widely used to identify non-native species and their source population in the native region [25,26].

mtDNA phylogenetic analysis of sika deer in mainland South Korea identified two subspecies of alien sika deer, *Cervus nippon taiouanus* and *Cervus nippon yesoensis* [15,27]. To the best of our knowledge, no molecular research has been performed concerning the taxonomic studies, evolutionary relationship of introduced sika deer, and their native geographical range in Jeju. Therefore, the maternal origin of the sika deer on Jeju Island remains unknown. Hence, this study aimed to determine the taxonomic and phylogenetic relationships of sika deer introduced to Jeju Island using mtDNA *CytB* gene sequences. This study aimed to resolve the taxonomic status of sika deer to understand their invasion of Jeju Island, South Korea, to support management strategies, including controlling and limiting their population in the wild.

## 2. Materials and Methods

### 2.1. Samples Collection and Laboratory Procedure

Nine sika deer tissue samples were obtained from roadkill (two samples) and the World Natural Heritage Headquarters (WNHH) in Jeju (seven samples), collected under the invasive deer population control program of the local government. Samples were collected from various locations on Jeju Island (Figure 1) and stored in centrifuge tubes containing 30 mL 95% ethanol. Total DNA was extracted using the DNeasy Blood and Tissue Kit (Qiagen, Germany) to a final volume of 80 μL and stored at 4 °C until further analysis.

mtDNA *CytB* marker was selected for molecular identification of sika deer. *CytB* sequences were amplified using newly designed primers: forward (5′-CAAGAACACTAATGACCAATATCC-3′) and reverse (5′-TACAAGACCAGTGT ATTGAG TAT-3′). Polymerase chain reactions (PCR) amplifications were performed using a MiniAmp Plus Thermal Cycler (Thermo Fisher Scientific, Waltham, MA, USA). A total volume of 20 μL contained 17 μL distilled water, 1 μL (10 pmol) forward and reverse primers each, and 1 μL template DNA in a premixed ready strip (FastMix Frenche™ PCR i-Taq, iNtRon Biotechnology, Seoul, Korea). The thermal cycling conditions consisted of an initial pre-denaturation for 3 min at 95 °C, followed by 35 cycles of denaturation for 30 s at 95 °C, annealing for 1 min at 50 °C, and elongation for 1 min at 72 °C, with a final extension for 5 min at 72 °C. The amplified results were visualized on 1% agarose gels to verify the PCR quality. QIAquick purification kit (Qiagen, Valencia, CA, USA) was used to purify PCR products, which were then sequenced on an ABI 3130xl Genetic Analyzer (Applied Biosystems, Foster City, CA, USA).

### 2.2. DNA Sequencing and Phylogeny

All DNA sequences were assessed by a similarity search using the National Center for Biotechnology Information (NCBI) databases and the basic local alignment search tool (BLAST); the most identical putative species were listed. Using the ClustalW program, multiple sequence alignments were performed using the *CytB* sequences generated in this study and reference sequences collected from the NCBI database [28]. Haplotype analysis of *CytB* gene sequences of sika deer was performed using DnaSP v5 [29]. A maximum likelihood (ML) phylogenetic tree was produced using the Mega-X program [30], and the Tamura–Nei model [31] with gamma distribution (T93+G) was selected. The Hasegawa–Kishino–Yano (HYK) model [32] was chosen as the best-fit nucleotide substitution model using the Akaike information criterion [33] applied in MrModeltest v2.3 (Uppsala University, Uppsala, Sweden) [34]. The tree reliability of the ML method was estimated using the bootstrap percentage after 10,000 replications. Two independent runs were used to assess convergence, and the first 25% of sampled trees were removed as ‘burn-in’. A 50% majority rule consensus tree was generated by running 100,000 generations of Markov chain Monte Carlo (MCMC) chains and sampling every 100 generations. The reliability of the inferred nodes was tested using posterior probabilities. Sequences of the Siberian roe deer (*Capreolus capreolus*) were used as an outgroup to root the phylogenetic tree. The average genetic distance between determined haplotypes and reference sequence from the NCBI was calculated using the Kimura-2-parameters model with Mega-X version. The time of divergence (t) was computed using a constant molecular clock as t = K/2μ, where ‘K’ is the proportion of nucleotide differences between two sequences and ‘μ’ is the rate of nucleotide substitution [35]. For the control region of the deer mtDNA, a divergence rate of 1.11–1.31% per million year (Myr) was used [36]. The complete mtDNA sequences of *CytB* were submitted to the NCBI database using the submission tool, and their accession numbers are listed in Table 1.

## 3. Results

Nine samples were collected from sika deer (*C. n. yakushimae*, n = 5; *C. n. taiouanus*, n = 4) on Jeju Island. All collected tissue samples were successfully amplified and sequenced to evaluate their phylogenetic relationships. Two distinct *Cyt*B haplotypes of sika deer were found in the nine *CytB* sequences obtained in this study, reflecting the two previously introduced populations of sika deer on Jeju Island. One haplotype (*Cyt*BG1) was found in five sequences identical to that in *C. n. yakushimae* reported from Japan (AB218689). The remaining four sequences were obtained as a single haplotype (*Cyt*BG2) identical to that of *C. n. taiouanus* reported in Taiwan (DQ985076) (Table 1). We estimated the genetic distance between the two haplotypes (*Cyt*BG1 and *Cyt*BG2) and the reference sequences obtained from the NCBI database. The lowest genetic distance (0.000) was obtained between *Cyt*BG1 and sequence AB218689 reported in Japan, strongly suggesting that the *Cyt*BG1 haplotype is *C. n. yakushimae*. Similarly, the haplotype *Cyt*BG2 and sequences DQ985076 from Taiwan showed the lowest genetic distance (0.000), indicating that the haplotype *Cyt*BG2 is *C. n. taiouanus* (Table 2).

To resolve the phylogenetic position of the two haplotypes, ML tree based on Tamura–Nei model were constructed, which produced a robust phylogenetic tree distinctly clustered into two clades (Figure 2). Among these clades, the haplotype of *C. n. yakushimae* clusters in *C. n. yakushimae* (AB218689) clade has been reported from Southern Japan. However, another haplotype of the *C. n. taiouanus* cluster in *C. n. taiouanus* clade was reported from Taiwan (DQ985076), the United Kingdom (L15083), and South Korea (GU377259). The lowest genetic distance from the Taiwan lineage indicated that the species was introduced from Taiwan.

Furthermore, tentative divergence times between the two haplotypes determined in this study (*Cyt*BG1 and *Cyt*BG2) were at least 1.81 Myr (Figure 2). However, the divergence time between *CytBG1* and *C. n. yakushimae* (Japan) and *CytBG2* and *C. n. taiouanus* (Taiwan) were less than 0.17 Myr, indicating recent divergence. 

## 4. Discussion

This study was designed to identify the sika deer present on Jeju Island. This study determined some salient results as follows: (1) molecular identification, phylogenetic relationship, and tentative divergence time of sika deer; (2) sympatric distribution of two subspecies of sika deer, *C. n. yakushimae* and *C. n. taiouanus*, are present on Jeju Island; (3) both *C. n. yakushimae* and *C. n. taiouanus* are alien species to Jeju Island introduced from Japan and Taiwan, respectively.

Molecular identification and phylogenetic analysis of non-native populations can provide precise information about their source population. The sika deer, a non-native species on Jeju Island, shows an increasing demographic trend [45]. Furthermore, the genetic validity of its subspecies and the true range of its geographical origin have yet to be determined. This study focused on the initial investigation to provide correct information on the classification of non-native sika deer species and their native geographical origin on Jeju Island. Our study revealed the presence of two highly distinct sub-species of sika deer, *C. n. yakushimae* and *C. n. taiouanus*, which are the same as those in Japan and Taiwan, respectively. To the best of our knowledge, this study is the first molecular phylogeographic study of non-native sika deer that sheds light on the evolutionary history of the species. 

Morphologically, the two subspecies, *C. n. yakushimae* and *C. n. taiouanus*, are indistinguishable because of their similar body coloration, antler and body shape, and size [46,47]. The pelage of both sika deer ranged from reddish-brown in summer to dark brown or black in winter. In summer, they have white spots and a dark dorsal stripe terminating in a large rump patch, which is used as a distinguishing characteristic. Males have strong and erect antlers, with an additional buttress protruding from the brow tine. Females carry two prominent black bumps on their forehead. Males have a dark, shaggy mane on their necks during the mating season. Because of the difficulty in distinguishing the two subspecies based on their morphological characteristics, molecular datasets were used to identify such sympatric species. Therefore, we confirmed both species are sympatrically distributed in the natural ecosystems of Jeju Island.

Our study determined two distinct *CytB* haplotypes (*Cyt*BG1 and *Cyt*BG2) that were used to compute the phylogeny of sika deer using ML tree (Figure 2). The phylogenetic tree clearly shows that the two haplotypes clustered into two different groups representing *C. n. yakushimae* and *C. n. taiouanus.* The zero genetic distance between the haplotypes CytBG1 and C. n. yakushimae and *Cyt*BG2 and *C. n. taiouanus* indicated no difference between each haplotype and the underlying gene sequence. The identical sequence present in each group indicated an exact match between the species [22,23,48]. In addition, genetic differences between closely related species typically result in minimal DNA sequence divergences [49]. Based on this evidence, we concluded that the haplotypes *Cyt*BG1 and *Cyt*BG2 identified on Jeju Island were *C. n. yakushimae* and *C. n. taiouanus,* respectively, and the source population might be from Japan and Taiwan.

In this study, we used tissue samples to obtain more qualitative DNA; however, the sample size was limited because all samples were obtained from roadkill and the WNHH under the deer management plan. However, we believe that this sample size is sufficient for the molecular identification of the species, following earlier studies [50,51]. 

The subspecies *C. n. yakushimae* is the most common and smallest subspecies of sika deer native on two small islands in the southeastern portion of Japan [39,52], and has also been documented in non-native areas in the United States and the United Kingdom [53,54]. The population of *C. n. yakushimae* in Japan has increased since the 1970s [55,56], and deer herbivory has caused severe damage to agricultural and forested lands owing to grazing and browsing pressure [52,57]. To mitigate further damage, the expanding population has been culled or translocated to other countries [47,58], as was the case with the sika deer introduction in South Korea. The sequences GU377259 from South Korea and DQ985076 from Taiwan showed higher similarity in BLAST search results, which was the *CytB* gene sequences of *C. n. taiouanus* living in Songnisan National Park, mainland South Korea. This result indicates that *C. n. taiouanus* from the same origin group in Taiwan was introduced to Songnisan National Park and Jeju Island. Another possibility is that sika deer were re-introduced to Jeju Island shortly after being introduced into Songnisan National Park, not having been inhabited long enough to have gone through the evolutionary process. Our study revealed that two subspecies of sika deer were re-introduced into Jeju Island from Taiwan and Japan to restore the sika deer population after the extinction of native species from Jeju Island.

The molecular date for the sika deer lineage was determined based on the divergence rate of 1.11–1.31% per Myr for the deer mtDNA regulatory region [36]. Our results showed that the common ancestor of these two clades diverged in Northeast mainland Asia during the early Pleistocene epoch (1.81 Myr), which corresponds approximately with the findings of Ba et al. [59] and Guo et al. [60]. Pleistocene sika deer expanded their range southward and eastward in two ways: one crossed the Korean Peninsula and migrated to the Japanese island via a land bridge at least twice [61] between the southern section of the Japanese island and the Korean Peninsula during the middle Pleistocene period. Similarly, the other way followed the mainland’s east coast, leading to Vietnam and Taiwan. Several earlier investigations have supported these scenarios [38,59,62].

The Dybowski sika deer (*C. n. hortulorum*) was once found throughout most far eastern Russia, north-eastern China, and the Korean Peninsula [63,64,65]. In South Korea, *C. n. hortulorum* is considered the largest native deer subspecies [66]; however, our study samples could not detect this species on Jeju Island. In addition, an earlier study could not report *C. n. hortulorum* in the deer farms of mainland Korea and Jeju Island [15]. This species seems to be either extinct in the wild due to poaching and overhunting or lack relevant information [27,67]. Hence, surveillance studies based on large sample sizes are necessary to confirm the status of the species.

Members of the genus Cervus often hybridize and produce fertile offspring in the regions where they were introduced [68,69,70]. Sika deer in Northeast Asia have experienced various gene flows such as migration, wild population introduction, and exchange between farms and conservation parks since the early 1900s [65,66]. However, no sequence variation was observed among sika subspecies in this study, and the chance of interbreeding in the near future cannot be neglected. Many groups of sika deer in Japan [71,72], the Czech Republic [64], Europe [73,74], and New Zealand [75] have experienced significant genetic pollution. The ecological consequences of this hybrid phenomenon cannot be anticipated. Intervening hybrids facilitate gene flow, resulting in positive feedback that eventually blends the two populations [76]. We plan to study the genetic hybridization of this species in the near future.

There are no natural predators, such as tigers, leopards, and jackals, on Jeju Island as in other countries; therefore, the deer population has increased on a large scale compared to previous reports [45]. However, there could be intra-competition among the sika deer and inter-competition with roe deer for forage and natural habitats [77]. The rapidly increasing sika deer population in Jeju has reduced the number of dietary plants of native roe deer, particularly in the winter season [77]. The government has started to control the sika deer population in Songnisan National Park (mainland South Korea) to reduce human conflict and long-term conservation [41]. On Jeju Island, sika deer have not yet impacted the environment seriously; however, some cases of road accidents and human conflicts have been recorded [78]. Therefore, regularly monitoring the deer population is required to study the behavioral patterns and habitat management of sika deer and other native species.

## 5. Conclusions

The phylogenetic analysis of non-native populations using molecular genetic methods can provide information on the exact species identification and source population. This first phylogeographic survey of sika deer on Jeju Island suggested the presence of two distinct subspecies (*C. n. yakushimae* and *C. n. taiouanus*) introduced from Japan and Taiwan. Although the sample size was small, we extracted valuable information regarding the taxonomy and phylogeny of the sika deer currently present on Jeju Island. Our study provides valuable data for identifying sika deer and their alien status. Our findings could be a reference for the government and conservation agencies for the long-term conservation of biodiversity on Jeju Island.

## Figures and Tables

**Figure 1 animals-12-00998-f001:**
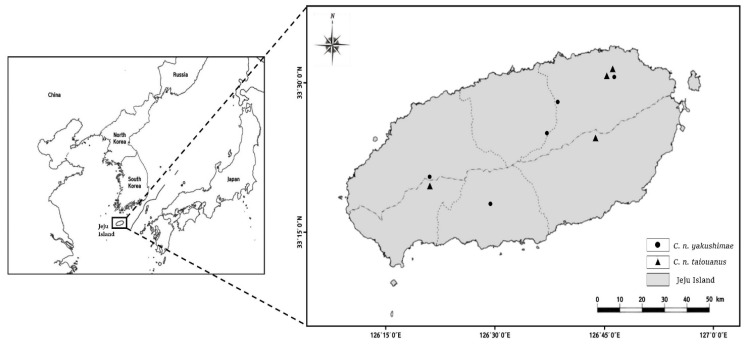
Map of Jeju Island showing samples collection sites.

**Figure 2 animals-12-00998-f002:**
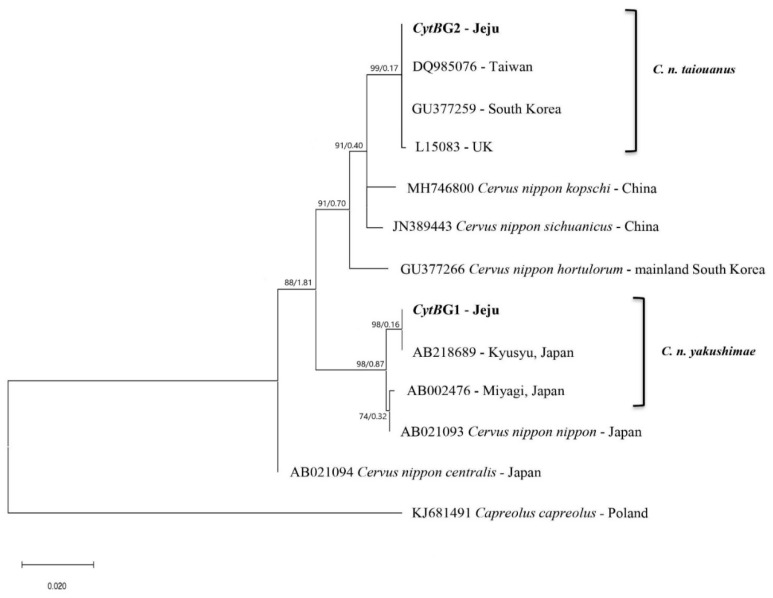
Maximum likelihood (ML) phylogenetic tree based on the mtDNA *CytB* gene sequences for two haplotypes of *C. nippon* collected from Jeju Island and reference sequences of various Cervids collected from the NCBI database. The numbers at each node represent bootstrap value and tentative divergence time, respectively. The tentative divergence time was estimated in million years. *CytB* gene sequences of 10 Cervid species were used as reference sequences and *Capreolus capreolus* was used as an outgroup. Detailed information of haplotypes and sequences determined in this study are presented in Table 1.

**Table 1 animals-12-00998-t001:** Accession numbers of *CytB* gene sequence and corresponding haplotypes of *C. nippon* used in this study.

Haplotype	Accession No.	Species	Origin Country	References
*CytBG1*	MW169432	*Cervus nippon yakushimae*	South Korea	This study
	MW169440	*Cervus nippon yakushimae*	South Korea	This study
	MW169449	*Cervus nippon yakushimae*	South Korea	This study
	MW169451	*Cervus nippon yakushimae*	South Korea	This study
	MW169452	*Cervus nippon yakushimae*	South Korea	This study
*CytBG2*	MW169437	*Cervus nippon taiouanus*	South Korea	This study
	MW169438	*Cervus nippon taiouanus*	South Korea	This study
	MW169444	*Cervus nippon taiouanus*	South Korea	This study
	MW169445	*Cervus nippon taiouanus*	South Korea	This study
	AB021094	*Cervus nippon centralis*	Japan	[35]
	AB002476	*Cervus nippon yakushimae*	Japan	[37]
	L15083	*Cervus nippon taiouanus*	UK	[38]
	AB021093	*Cervus nippon nippon*	Japan	[35]
	AB218689	*Cervus nippon yakushimae*	Japan	[39]
	DQ985076	*Cervus nippon taiouanus*	Taiwan	[40]
	GU377259	*Cervus nippon taiouanus*	South Korea	[41]
	GU377266	*Cervus nippon hortulorum*	South Korea	[42]
	JN389443	*Cervus nippon sichuanicus*	China	[43]
	MH746800	*Cervus nippon kopschi*	China	[44]
	AB021094	*Cervus nippon centralis*	Japan	[35]

Note: Species name of the *CytB* gene sequences used in this study were determined from a BLAST search of the NCBI database.

**Table 2 animals-12-00998-t002:** Pairwise genetic distance between the haplotypes of *C. nippon*.

Haplotype/Accession No.	1	2	3	4	5	6	7	8	9	10	11	12
1. CytBG1—Jeju												
2. CytBG2—Jeju	0.0481											
3. L15083—UK	0.0499	0.0018										
4. AB218689—Japan	**0.0000**	0.0481	0.0499									
5. AB002476—Japan	0.0093	0.0500	0.0517	0.0093								
6. GU377259—Korea	0.0490	0.0010	0.0028	0.0491	0.0509							
7. DQ985076—Taiwan	0.0481	**0.0000**	0.0018	0.0481	0.0500	0.0010						
8. MH746800—China	0.0463	0.0178	0.0196	0.0463	0.0482	0.0187	0.0178					
9. AB021094—Japan	0.0334	0.0346	0.0364	0.0334	0.0352	0.0356	0.0346	0.0329				
10. GU377266—Korea	0.0432	0.0253	0.0271	0.0432	0.0451	0.0263	0.0253	0.0236	0.0298			
11. JN389443—China	0.0437	0.0152	0.0170	0.0437	0.0456	0.0162	0.0152	0.0134	0.0303	0.0118		
12. AB021093—Japan	0.0046	0.0384	0.0457	0.0053	0.0028	0.0373	0.0363	0.0405	0.0385	0.0315	0.0046	

Note: CytBG, Cytochrome B Group; UK, United Kingdom. Values in bold represent lowest genetic distance between pairs.

## Data Availability

The data generated and analyzed during this study are available in the Genbank repository (Accession Nos. MW169432–MW169445).

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
