# Peer review of "Molecular Evidence Reveals the Sympatric Distribution of Cervus nippon yakushimae and Cervus nippon taiouanus on Jeju Island, South Korea"

_animals, 2022, doi:10.3390/ani12080998_

Round 1

Reviewer 1 Report

I am sorry, but I still cannot recommend publication.

1. I still do not understand what is going on in Figure 2.

  1. A) How can it be BOTH an NJ and an ML tree?

  1. B) If this is true, then why are both trees superimposed on the other and the same cladistic morphology? How are these trees the same shape and branch length?

  1. C) I am just as confused about the bootstrap values for the same reasons. The third value is the tentative divergence time, but there are no units given.

  1. D) Finally, I STILL do not understand how to apply the time-line under the tree to the data. You say “major relevant epochs are shown below the phylogenetic tree.” However, is that supposed to be employed with the divergence times to the tree itself? Why is it below the tree at all?

  1. E) BTW, the term “phylogram” is never used in the context of this tree—even though it is a phylogram.

2) Did you really only use TWO markers to do this study? Two markers AND just only NINE deer? If this is true, then I am missing the basic validity of this work. My understanding has always been that a minimum of six-seven polymorphic markers are necessary for any type of phylogenetic study to have statistical relevance. Am I missing something here? Please explain.

Author Response

Manuscript ID: animals-1562356.

Reviewer #1:

We are grateful to reviewer 1 for your valuable comments and suggestions for the qualitative improvement of our manuscript. Here we are responding to each question and comment. Please find the point-to-point response below.

Q 1. I still do not understand what is going on in Figure 2.

  1. A) How can it be BOTH an NJ and an ML tree?

Response: Considering the comments and suggestions of all reviewers, we revised Figure 2 and removed the NJ tree in our revised manuscript.

  1. B) If this is true, then why are both trees superimposed on the other and the same cladistic morphology? How are these trees the same shape and branch length?

Response: We revised Fig. 2 please refer to Fig. 2.

  1. C) I am just as confused about the bootstrap values for the same reasons. The third value is the tentative divergence time, but there are no units given.

Response: We hereby presented the bootstrap value only for the ML tree.  The unit for divergence time was million year (Myr) which was not mentioned in the Figure legend. We corrected it in our revised manuscript.

  1. D) Finally, I STILL do not understand how to apply the timeline under the tree to the data. You say “major relevant epochs are shown below the phylogenetic tree.” However, is that supposed to be employed with the divergence times to the tree itself? Why is it below the tree at all?

Response: The timeline present underneath the phylogenetic tree was removed. The legend of Figure 2 was also revised and presented in our revised manuscript.

  1. E) BTW, the term “phylogram” is never used in the context of this tree—even though it is a phylogram.

Response: Thank you for your suggestion. The term “Phylogram” has been replaced by the term “phylogenetic tree”. Please refer to lines 269-270.

Q 2.  Did you really only use TWO markers to do this study? Two markers AND just only NINE deer? If this is true, then I am missing the basic validity of this work. My understanding has always been that a minimum of six-seven polymorphic markers are necessary for any type of phylogenetic study to have statistical relevance. Am I missing something here? Please explain.

Response:

The main objectives of our study were molecular identification and phylogenetic analysis of sika deer present in Jeju Island. We used nine complete mitochondrial DNA Cytochrome B (CytB) gene sequences, at which two CytB haplotypes were determined. These two haplotypes are identical to Cervus nippon yakushimae native sika to Japan and Cervus nippon taiouanus native to Taiwan. As similar to us Qu et al. (2019) and Li et al. (2020), showed a phylogenetic study using small sample size.

References:

  • Qu et al., 2020. Minimum sample sizes for invasion genomics: Empirical investigation in an invasive whitefly. Evol. 2020, 10, 38–49.
  • Li, et al., Optimizing Sample Size for Population Genomic Study in a Global Invasive Lady Beetle, Harmonia Axyridis. Insects 2020, 11, 290.

Thank you

Reviewer 2 Report

Dear Editor,

I reviewed the revised version of the manuscript: “Molecular evidence reveals the sympatric distribution of Cervus nippon yakushimae and Cervus nippon taiouanus on Jeju Island, South Korea”.

The authors roughly answered the questions raised. However, the manuscript is improved in this new version so I endorse its publication in its current form.

Author Response

We are grateful to the reviewer for your valuable comments and suggestions for the qualitative improvement of our manuscript. In accordance with your suggestion, we made some corrections in language and in the manuscript section. 

This manuscript is a resubmission of an earlier submission. The following is a list of the peer review reports and author responses from that submission.

Round 1

Reviewer 1 Report

Banjade et al.’s manuscript on the population genetics sika deer on Jeju Island has some important points on the consequences of introducing non-native species to an isolated area. Its science is mostly well thought out and its experiments and conclusions seem to be well-documented. However, I have a number of problems with the paper that prevent publication.

1) On the science side of things, I am concerned because the paper employs a total N=9 deer. It may be because I deal with population genetics of plants more than animals, but that seems like a very low number of samples. Is that normal? Please defend this as a common process in the field of animal genetics.

2) In my own work, we tend to employ upwards of 15-20 polymorphic alleles to analyze populations of 30-40 individuals. I am a bit disturbed by the analysis of just two haplotypes—even if it is mitochondrial sequence. Again, please defend this as a common practice in your field.

3) The time line in Figure 1 is very confusing. Please place it in such a way that it is clear as to what exactly you are referring.

4) I missed your explanation for why you decided on the use of a ML tree as opposed to the other myriad types.

5) I strongly urge you to get a Native English speaker to edit your paper before you resubmit it. There are many issues with redundancy, misused articles, misused verbs, and non-standard sentence construction. The paper needs a serious overhaul as far as the writing goes. Just a few examples:

                Line 20 “…particularly on island…”

                Line 30 “…are currently exist…” Do you mean “…currently exist”?

                Line 35 “…by introducing non-native species.”

                Line 38 “The impacts of non-native species have been evaluated typically more severe…”

                Line 76 “This study expects to solve taxonomic controversies…” Well? Does it? Do you mean "“This study solves taxonomic controversies…” ?

                Line 238 “This study could reference the government and conservation agencies…” Do you mean “...could be cited by…”?

6) Line 54 “Although there is scant evidence of introduction, the population of sika deer is growing faster…” This statement makes no sense since the entire paper is about the history of the introduction of the species. How can you state there is scant evidence? The sika deer are living there! Isn’t their continued existence on the island show us that they were introduced? If you mean something else, then please clarify.

7) Finally, why do the authors believe that data availability is “not applicable”? There is sequence data generated by this study, so it should be available for general enlightenment.

Reviewer 2 Report

The paper by Banjade and co-authors reports the presence of two subspecies of Cervus nippon in Jeju island, based on the sequencing and phylogenetic analysis (with comparison with previously generated sequences available at NCBI - GenBank) of the mtDNA of nine deer samples. The paper is overall well presented, but there are some issues with the sue of english languege. While not being a native speaker myself, I made some suggestions troughout the manuscript, that I send in the paper version attached to this review. While the authors highlight the importance of the study for it helps clarify a dispute on the origin of the deers found in the island, the dispute is not described with adequate detail. I do not question that such dispute does occur but it would be useful to go a bit deeper into that. The authors use only nine samples, without referring to where these were collected in Jeju island. No sampling locations are provided nor results (distinct haplotypes) are presented by location. This is important to assess whether the subspecies are spatially segregated in the island or not. It would also be intersting to have a larger number of samples but the fact is that the nine samples were enough to  confirm the presence of distinct mtDNA lineages consistent with a two-subspecies scenario. The authors refer to hybridization, which is a very likely scenario in close cervid subspecies, but their study design does not allow them to test it. Testing for hybridization (which demands more samples and different - polymorphic and recombinant nuclear - markers, would greatly improve the manuscript but I imagine it is hard to accomplish it at this moment. Beside the small suggestions I made in the attached file, I also send some additional comments below.

Additional Comments

Abstract

Line 21 – “will aid in their prevention”: prevention of what? Biological invasions? It is not very clear, so please consider rephrasing.

Lines 25-26 – Consider rephrasing to: “revealing two distinct groups of CytB haplotypes: Cervus nippon yakushimae, a sika deer subspecies native to Japan; and Cervus nippon taiouanus, a sika deer subspecies native to Taiwan.”

Lines 26-27 – Consider rephrasing to: “These results showed that two subspecies of sika deer, native to Japan and Taiwan, were introduced in Jeju Island, South Korea”

Lines 28-31 – There is some redundancy with the rest of the abstract. Please consider rephrasing, including additional information, or summarize these last sentences in order to avoid redundancy.

Introduction

Lines 45-46 – if you are not specifically speaking of islands, in this case, it would be important to consider non-intentional invasion such as the one caused by animals that escape from captivity or that disseminate autonomously from an original source of introduction. That is often common in Europe, with carnivore species that escaped farms for fur production in Central and Eastern Europe and are currently colonizing new countries. Even in islands, such non-intentional mammal invasions, such as from rodents, may occur in islands. Please consider refering to non-intentional introductions of invasive mammal species or else clarify the context for referring only to intentional introductions, if are you referring the specific case of ungulates in islands?

Methods

Lines 82-84 – In the abstract you mention roadkill and samples from the World Natural Heritage Headquarters but here you do not mention roadkill. How many samples were collected from roadkills and how many samples were obtained from the World Natural Heritage? Do you have any spatial data (GPS coordinates or sampling locality) for these samples?

Lines 89-91 – Were the primers developed by your team or were previously designed by other authors? If they were designed by you maybe it could be useful for other authors to include a designation for the primers. If they were designed by other authors, please cite the original study where they were described first and refer to their designation in the original study.

Lines 107-109 – Which software did you use to construct the ML tree? How did you selected the best substitution model? I can see that you selected the model and parameters using AIC but it is not clear which software did you use or which alternative did you use…

Lines 111-113 – Did you included a burnin period or did you otherwise discarded the first random walk steps? 10,000 replications seems little for a ML procedure but it is not an impediment… however, in this case, it would be important to test for convergence. Did you conducted several independent runs and tested for convergence?

Lines 117-118 – are you providing reference [34] as a reference to the procedure for estimating the molecular clock or for the value you used for the rate of nucleotide substitution. I think the authors of this paper refer to Kuyawama and Ozawa (2000) for the estimation of the substation rate. Please be sure that credits are attributed to original authors as well. And also include the nucleotide substitution rate you used.

Results and Discussion

In first place, why didn’t you divided results and discussion in two distinct sections. Results are not complex to present and I see no evidence that prior discussion of a result was necessary for deciding the analysis strategy to get some other results. I think it would be easier for you to convey (and for readers to follow) your story, if you clearly separate (in two sections) your results from the discussion of your results and the ones from other authors.

Lines 122-123 – you provide no map indicating sampling locations (nor geographical coordinates of these locations. Given the fact that two subspecies were identified in the island, it would be important to know if these occur in sympatry or if they occupy distinct areas of the island. Could you please provide a map with the location of 3each sample and, preferably, depicting (e.g. through distinct colors or shades of grey) the sequenced haplotype, in each case…

Lines 124-125 – actually, Kubo & Takatsuki indicate that morphological variation in the species is consistent with Bergmann’s rule. Taiwan is located much southern than Japan and South Korea. Differences on body size should be expected in both subspecies, unless phenotypic variation is more a result of ecological plasticity than a result of genetic differences. Also, differences could be diluted with hybridization, if this is likely to occur among subspecies. Did you account for the possibility of hybridization among the two subspecies. This is important for mitochondrial DNA only provides information on the matrilineal history and is not able to catch the sign of hybridization, when considered alone. Are the two subspecies known to hybridize? Did you accounted for that possibility?

Lines 141-142 – the four samples were treated as a single haplotype for it were the same haplotype (for the sequenced fragment)? If so, please simplify: “ the remaining four sampes (…) shared the same haplotype, that was identical to one reported in Taiwan”. If not, please clarify how and why you pooled these samples in a single haplotype.

Lines 148-149 – if the sequences are identical (by state, at least) it is expected that genetic distance would be 0… in fact, the explanation in the next lines is spurious for there is no need of explaining which is the genetic distance among two haplotypes when they are identical (by state and, assumingly, by descent).

Figure 1 – Please make it clearer in the caption which is the exact meaning of both figures placed at every node. Is it the time scale and the support for node? If so, in each unit is given the time at each node? And the node support is a bootstrap, a likelihood, a posterior probability?

Additionally, in your tree, Cervus elaphus is not reciprocally monophyletic to C. nippon. Were you expecting this result? Why? Or was it surprising? If so, what do you think of this result.

Lines 170-171 – What do you mean when you refer to “a robust and identical phylogenetic tree (…)”? Identical to what? Please be aware that your tree does not show a clear split in two clades. Even without knowing which is the parameter you used for the level of support for nodes, in the tree in figure one, you have:

  • A clade with 96 support, for n. taiouanus and other C. nippon subspecies;
  • An unclear clustering of elaphus, among other C. nippon haplotypes.
  • A clade with only 91 support (low if the parameter is probability or posterior probability) for all nippon yakushimae and higher support only for two of the haplotypes of this species, including the one sampled by you in Jeju.

Table 2 – haplotypes in this table are not coded using the accession numbers as in Table 1 and Figure 1. That makes it impossible to track which haplotype code stands for which genebank accession numbers. Please standardize the designation of haplotypes so it can be feasible for readers to follow your line of reasoning.

Lines 184-187 – Which sequences are those? Where are they presented in your paper. Are those the sequences retrieved from the blast? If so, please rephrase and clarify. I could not understand what you wished to convey here.

Lines 190-191 – Since you are discussing the time since colonization of islands as a way to support the low likelihood of evolutionary (genetic) divergence, it would be interesting to include some information on the likely date of each of the introductions you are discussing here, either on Jeju, or in other locations that you mention.

Lines 195-197 – What do you mean here? In what way, the presence of the two subspecies in Jeju suggests the existence of several factors that shape the patterns of geographical distribution of this species? Please clarify…

Lines 225-226 – mtDNA chromosome is haploid in mammals (with exception to some heteroplasmic individuals) and is not recombinant. Even if hybridization does occur in Jeju, you wouldn’t be able to track it genetically only using mtDNA. You would have to genotype or sequence nuclear (recombinant) markers in order to detect hybridization. The lack of variation on the genetic sequences is not evidence of absence of hybridization among populations or subspecies. You simply did not test for hybridization and your study design does not allow for assessing it. Please rephrase. If you aim to discuss hybridization here, you should at least discuss why your study does not allow for testing hybridization. You also should go deeper on how likely it is for the two subspecies to hybridize (with reports from other regions of the world).

Reviewer 3 Report

Dear Editors,

I regret to inform you that I recommend this manuscript for major revisions. It contains fundamental errors that have to be rectified through author revisions. I enclosed some fundamental comments below.

The paper describes the first molecular evidence of the presence of Cervus nippon yakushimae and Cervus nippon taiouanus in Jeju Island, South Korea.

Major revision:

The authors collected nine samples from roadkill and from the World Natural Heritage Headquarter in Jeju and, by analyzing the CytB gene, they established their belonging to the two subspecies Cervus nippon yakushimae and Cervus nippon taiouanus, introduced in the past in Korea. The number of samples is so low, and it does not allow to exclude the existence of other subspecies.

The authors should analyze more individuals with more sequence markers. Nuclear genomic markers (i.e. microsatellites) should be included to better characterize the sika deer Cervus nippon yakushimae and Cervus nippon taiouanus in Jeju. In this way the phylogenetic analysis will also be improved.

Minor revision:                                           

1-The haplotype CytBG1 includes five samples identical to the gene sequences of C. n. yakushimae reported in Japan (AB218689); while the haplotype CytBG2 includes four samples identical to the C. n. taiouanus reported in Taiwan (DQ985076).

In both cases, the gene sequences are identical each other. Why do the authors use distinct accession numbers? (MW169432, MW169440, MW169449, MW169451 and MW169452 for CytBG1; MW169437, MW169438, MW169444, and MW169445 for CytBG2)

2-Since CytBG1 and AB218689 sequences as well as CytBG2 and DQ985076 are perfectly identical to each other, the estimate of their genetic distance is ridiculous! It is obvious that their genetic distance is zero! The authors must consider as once sequence the CytBG1 and  AB218689 as well as the CytBG2 and DQ985076 haplotype sequences

3-Likewise, identical sequences do not need to be used in the ML tree. It is obvious that the haplotype CytBG1 clusters in the clade of C. n. yakushimae (AB218689), and the haplotype CytBG2 clusters in the clade of C. n. taiouanus (DQ985076). They are identical sequences and identify a same branch in the tree!

4-The sequences presented in Tables and in Figure 1 are not the same. There is a bit of confusion. For instance, the species of C. n used for the phylogenetic tree are not described in the text. A brief description of them might contribute to understand the analysis.

Line 185- 191 should be revised:

-EF139156 and EF058308 are not inserted in Tables as well as in Figure 1.

-It is not clear to me the relationship between Formosan sika deer and C. taiouanus species. I understand that they are the same.

Line 198-200

-The relevant node is not indicated in the figure.

-Between which species was the divergence rate for the mtDNA control region calculated? The description of the timetree should be improved.

Line 209- 213

-If the the haplotype CytBG1 and AB218689 sequences and the haplotype CytBG2 and DQ985076 sequences are identical, why do you refer to divergence time?

Line 214-220

In table 1, C. n. dybowski e C. n. hortulorum are reported as two different species. Are they?
